# The evolution of organic material on Asteroid 162173 Ryugu and its delivery to Earth

H. G. Changela [1,2] ✉, Y. Kebukawa [3,16], L. Petera[1,4], M. Ferus[1], E. Chatzitheodoridis[5,6], L. Nejdl [7], R. Nebel [1], V. Protiva[1], P. Krepelka[8], J. Moravcova[8], R. Holbova[8], Z. Hlavenkova[8], T. Samoril[9,10], J. C. Bridges[11], S. Yamashita[12], Y. Takahashi [13], T. Yada [14], A. Nakato [14], K. Sobotkova[10], H. Tesarova[10] & D. Zapotok[15]

The recent return of samples from asteroid 162173 Ryugu provides a first insight into early Solar System prebiotic evolution from known planetary bodies. Ryugu's samples are CI chondrite-like, rich in water and organic material, and primarily composed of phyllosilicate. This phyllosilicate surrounds micron to submicron macromolecular organic particles known as insoluble organic matter. Using advanced microscopy techniques on Hayabusa-2 samples, we find that aqueous alteration on Ryugu produced organic particles richer in aromatics compared to less altered carbonaceous chondrites. This challenges the view that aromatic-rich organic matter formed pre-accretion. Additionally, widespread diffuse organic material occurs in phyllosilicate more aliphatic-, carboxylic-rich, and aromatic-poor than the discrete organic particles, likely preserving the soluble organic material. Some organic particles evolved to encapsulate phyllosilicate, indicating that aqueous alteration on Ryugu led to the containment of soluble organic matter within these particles. Earth therefore has been, and continues to be, delivered micron-sized polymeric organic objects containing biologically relevant molecules.

Prebiotic chemistry on the terrestrial planets such as early-Venus, -Earth, and -Mars may have led to first life in the Solar System. Comets[1], asteroids[2], Kuiper Belt Objects[3] and satellites[4] additionally record earlier planetary formation and prebiotic evolution, shedding light on physicochemical organic processes prior to the occurrence of life, at least on Earth. Although prebiotic chemistry on early planetesimals may not have bridged with biology, understanding why this bridge was not met is vital. Planetesimals from the early Solar System such as

[1]Department of Spectroscopy, J. Heyrovsky Institute of Physical Chemistry, Czech Academy of Sciences, Prague, Czechia. [2]Department of Earth & Planetary Sciences, University of New Mexico, Albuquerque, NM, USA. [3]Department of Chemistry and Life Science, Yokohama National University, Yokohama, Japan. [4]Department of Inorganic Chemistry, Faculty of Science, Charles University, Prague, Czechia. [5]School of Mining and Metallurgical Engineering, National Technical University of Athens, Athens, Greece. [6]ESTEC, European Space Agency, Noordwijk, The Netherlands. [7]Department of Chemistry and Biochemistry, Mendel University, Brno, Czechia. [8]Central European Institute of Technology Masaryk University, Brno, Czechia. [9]Central European Institute of Technology, Brno University of Technology, Brno, Czechia. [10]TESCAN GROUP a.s., Brno, Czechia. [11]Space Park Leicester, School of Physics & Astronomy, University of Leicester, Leicester, UK. [12]Institute of Materials Structure Science, High-Energy Accelerator Research Organization, Ibaraki, Japan. [13]Department of Earth and Planetary Science, The University of Tokyo, Tokyo, Japan. [14]Astromaterials Science Research Group, Institute of Space and Astronautical Science, Japan Aerospace Exploration Agency, Kanagawa, Japan. [15]TESCAN USA Inc, Warrendale, PA, USA. [16]Present address: Department of Earth and Planetary Sciences, Tokyo Institute of Technology, Tokyo, Japan. ✉e-mail: changela@unm.edu

asteroids – the building blocks of planets – are remnants of the formation of the proto planetary disk, recording astrophysical processes ranging from supernovae, interstellar space, the solar nebular to planetary bodies[5]. Their impact-gardening on early planetary surfaces superseded the eventual occurrence of life as understood on Earth[6]. Unravelling hydrothermal and organic physicochemical evolution between interstellar, nebular and early planetary stages will therefore provide a clearer account of prebiotic evolution and life's origins.

The *Hayabusa-2* mission successfully returned ~5 g of sub mm- to cm- sized fragments of Asteroid 162173 Ryugu (Ryugu here on in) for the first time from a carbonaceous asteroid. Carbonaceous asteroids are the most abundant type of asteroid in the Solar System, characterised by remote sensing as planetary bodies with among the lowest albedos and displaying optical/infrared hydration spectral features[7]. Two sites were sampled by touch-and-go (TAG) on the equator of Ryugu, the first from regolith and the second targeted at crater material ejected at lower depths by a ballistic projectile[8]. Samples from both TAG sites on the equator of Ryugu are of the most hydrated and organic-rich carbonaceous chondrite (CC) composition – CI (Ivuna type) (Fig. 1). Aqueous alteration of primary components pervades CI chondrites. All CI chondrites are petrologic type 1[9,10], lacking any distinct chondrules and Calcium Aluminium Inclusions (CAIs), consisting mainly of water derived secondary minerals including phyllosilicate, (Fe, Ni) sulphides, Fe-oxides, carbonates and phosphates[11]. They are however among the least abundant CC group in Earth's meteorite record[12]. That said, Ryugu's CI composition does not seem unique to this asteroid. More recently in September 2023, the *OSIRIS REx* spacecraft successfully returned samples from Asteroid 101955 Bennu (Bennu here on in). So far, ~120 g of Bennu have been curated from the sample return capsule where TAG was performed in a region north of Bennu called 'Nightingale.' Preliminary reports on Bennu grains show dark CI material similar to Ryugu; abundant in phyllosilicate, magnetite framboids, Fe Sulphides, carbonate and phosphates (NASA Press conference). Both of these carbonaceous asteroids are spinning top rubble piles, drawn into the inner Solar System probably from the main belt[13]. If Ryugu and Bennu are representative major carriers of carbonaceous asteroid material reaching Earth, the scarcity of CI chondrites in Earth's meteorite collection might be attributed to their relative fragility compared to other carbonaceous groups, possibly hindering their survival upon atmospheric entry.

Prior to the return of asteroid Ryugu and Bennu samples by the *Hayabusa-2* and *OSIRIS-REx* spacecrafts respectively, CCs have provided the most accurate record of early Solar System organic evolution. Studies of soluble organic matter (SOM) and insoluble organic matter (IOM) bulk meteorite separates coupled with the characterisation of OM in situ provide a comprehensive account of OM[14,15]. Carbonaceous chondrites have the highest concentrations of both SOM and IOM with up to ~4 wt % total organic carbon (TOC)[16]. Generally, hydration in the CCs positively correlates with the abundance of TOC. In the most hydrated CCs such as CI (Ivuna-type), CR (Renazzo-type), CM (Mighei-type) chondrite groups, although their organic mass fractions are minor to trace in bulk[17], their volumetric distributions in their matrices are ubiquitous on the nano-scale[18]. Micron to submicron organic particles (OPs)[19] consistent with the composition of IOM occur in primitive chondrite matrices. Insoluble organic matter is acid insoluble, solid macromolecular organic material[20] that is largely made up of small aromatic groups with short, highly branched aliphatic moieties forming side chains on and cross links between the aromatic moieties[21]. They are often individual rounded submicron objects, so-called nanoglobules[22], but also occur as dendrites or veins in CC matrix, featuring sub-particle nanoglobule morphologies[23] when imaged by scanning electron microscopy (SEM) and transmission electron microscopy (TEM). Here, we refer to a discrete organic object surrounded by inorganic material as an OP[19].

In the CCs, macromolecular OPs are mostly surrounded by either submicron (Fe, Mg, Al) hydrated amorphous silicate or phyllosilicate grains, and often finer Fe, Ni sulphide grains[18]. Carbon X-ray Absorption Near Edge Structure (XANES) analyses by Scanning Transmission X-ray Microscopy (STXM) shows two distinct spectral types of OP: the first is aromatic/olefinic (C = C)-carbonyl (C-O)-carboxylic/ester (COOR), referred to here as (C = C)-(C = O)-(COOR) '3-peak' OPs, which is characteristic of most CC IOM. The second is (C = C)-(COOR) '2-peak' OPs, aromatic-richer than 3-peak particles and lacking the carbonyl peak by XANES[19,24]. De Gregorio et al.[24] referred to these 2-peak aromatic richer OPs as 'aromatic' particles. Preliminary examination of OM in Ryugu samples by C-XANES[25] however referred to these 2-peak OPs

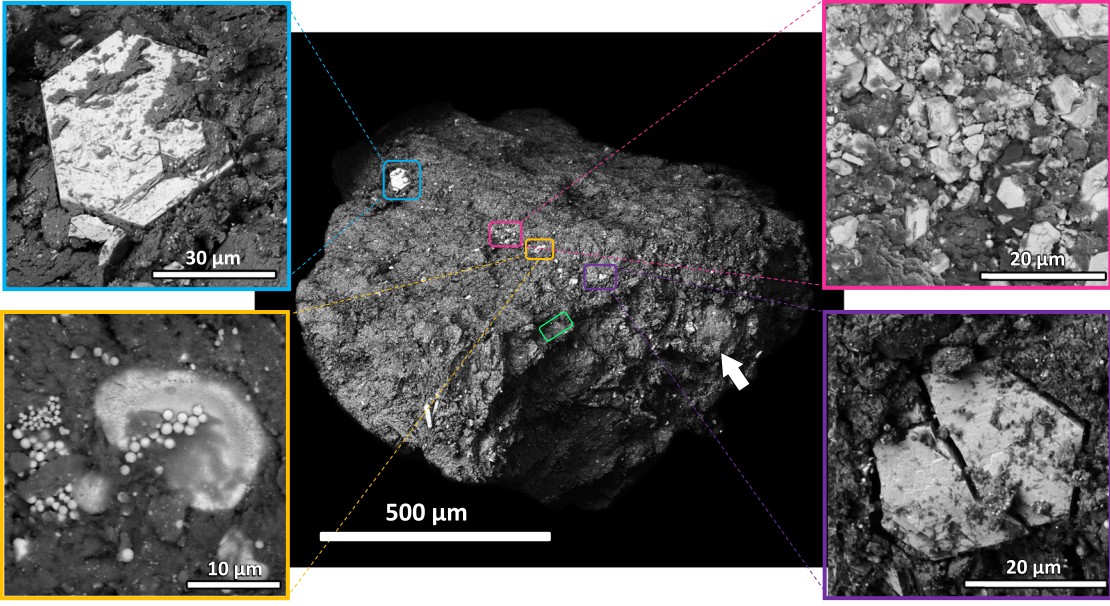

**Fig. 1 | Back Scattered Electron (BSE) images of particle A0083 (Radegast).** The central image is the entire image of Radegast. Insets show various phases on the top surface of the grain. Blue and purple borders are Fe-Sulphides (probably hexagonal pyrrhotite). Yellow is Fe-oxides (magnetite). Pink is a mixture of minerals. The green rectangle shows the region of tomography from the top edge of Radegast. White arrow shows an ~150 μm spherical nodule protruding from the side of the grain.

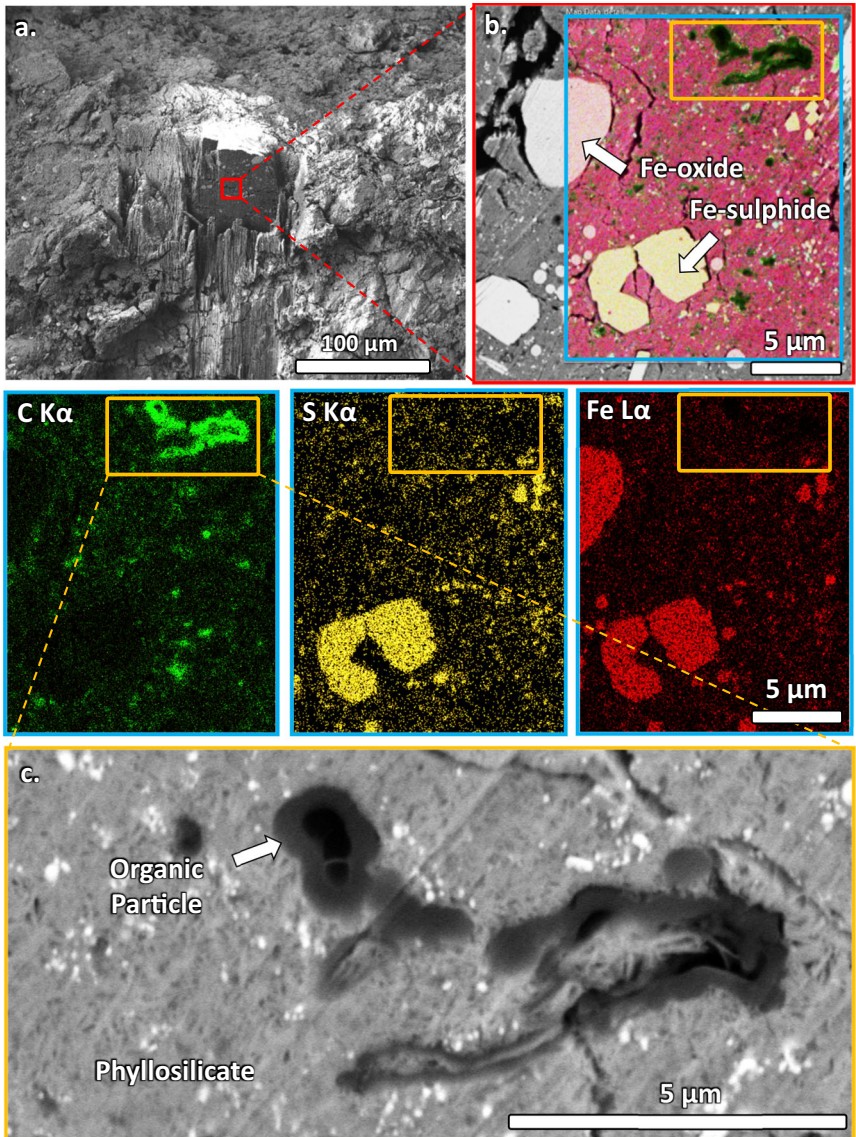

**Fig. 2 | Region of 3D tomography from the edge of A0083 and lamella extractions at the end of the tomography volume. a** BSE image of the edge of Radegast (green rectangle in Fig. 1) side on orientation. The region marked 3D is a milled FIB tomography region. **b** BSE and overlayed region of EDX map on the 1st cross section, with associated C Kα, S Kα and Fe Kα maps. Intense Fe Kα regions correspond to Fe-oxides (magnetite) and Fe Sulphide grains (intense yellow) from the S Kα map. Highest C Kα counts in the green map correspond to OPs widely distributed across the cross section. **c** Higher magnification BSE image of the coarse grained OP (white arrow and dark grey in BSE). Finer bright grains are mostly Fe Sulphides. The light grey surrounding groundmass is phyllosilicate.

as 'highly aromatic.' These aromatic-richer 2-peak OPs are less abundant than the 3-peak OPs in both IOM residues and CC matrices[19,24]. Based on previous studies, aromatic-rich 3-peak OPs are more abundant in the type 1 CR, GRO 95577[19] and the CI chondrite Orgueil[18] than in type 3/2 CCs, displaying a less intense carbonyl peak relative to the aromatic/olefinic peak. The same type of C-XANES is found in the 'aromatic particles' described in Yabuta et al.[25]. The aromatic/carbonyl peak ratios of these 3-peak OPs are higher than typical 'IOM-like'[25] 3-peak particles that make up the major fraction of IOM type 3/2 CCs, i.e., they are aromatic-rich. This means that other type 1 CCs such as the CR chondrite GRO 95577, which contains altered relic chondrules characteristic of its CC group, also display closer similarities with OPs from Ryugu, i.e., alteration probably led to their similar aromatic-rich compositions. This challenges the view that aromatic-richer IOM in Ryugu and type 1 CCs might be pre-accretionary OM[19,24,25]. The origin of chondritic IOM still remains controversial though[26]. Interstellar space[25,27], protoplanetary discs[22,25] and asteroids themselves[23,28] have

all been interpreted as settings under which the organic phases retained in carbonaceous asteroids have originated[29], possibly in non-mutually exclusive ways.

Hydrated silicates in CCs also ubiquitously bear a distinct type of OM – Diffuse OM[18]. Diffuse OM is widespread material, aromatic-poorer and carboxylic-richer than OPs, found within regions of phyllosilicate (phyllosilicate mostly occurs in type 1 CCs) and hydrated amorphous silicates (mostly found in higher petrologic type CCs up to type 3)[18]. Soluble organic material such as mono carboxylic and amino acids have been well documented in CCs[14,30] and in Ryugu samples[31,32]. The distribution of SOM is however poorly understood as its characterisation in situ is less developed. It most likely exists within hydrated phases such as phyllosilicate sheet layers and hydrated amorphous silicate grains[18], and occurs as coating around fibres of phyllosilicate and secondary minerals. Diffuse OM is more concentrated in type 1 than in type 2/3 CCs[18,19]. Soluble organic concentrations are however not enriched in type 1 CCs, although IOM

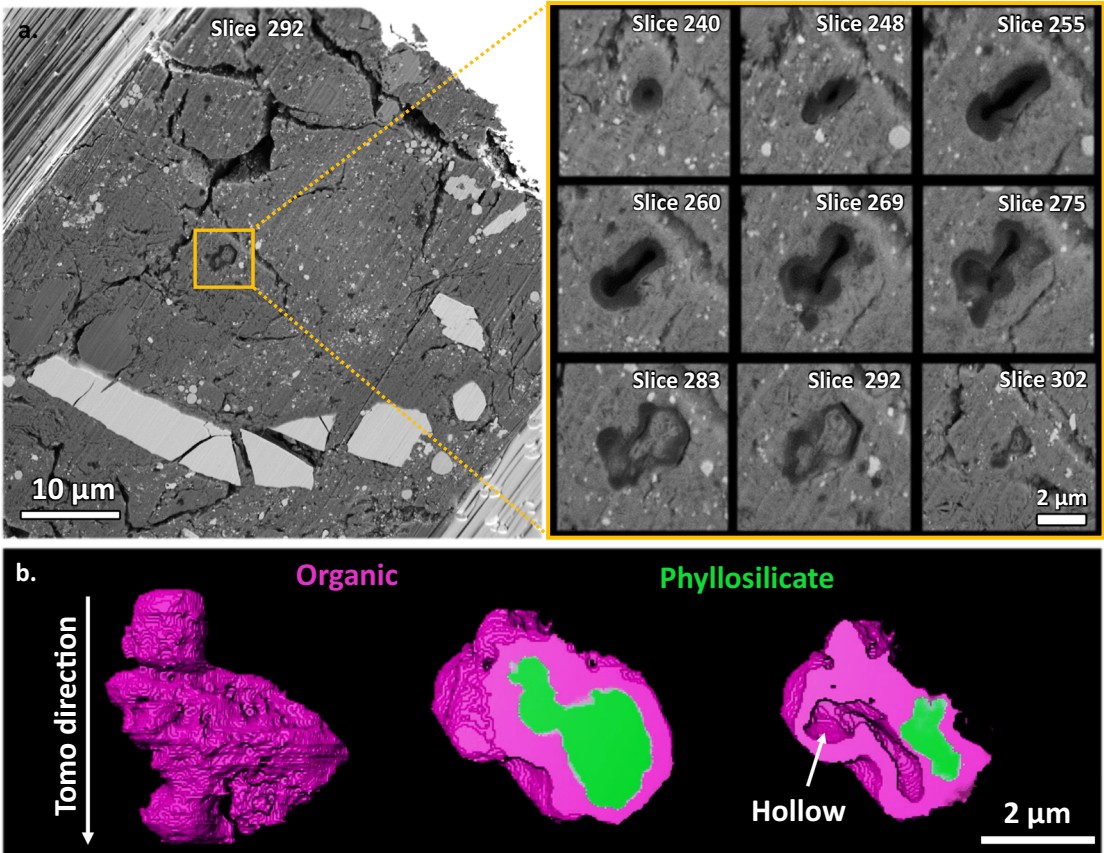

**Fig. 3 | Tomography of the coarse grained OP. a** BSE image of the entire tomography slice 292. EDX mapping (Fig. 4) shows the brightest phase in the BSE image as Fe-Suphide (probably plated pyrrhotite). Slightly darker grey are Fe-oxides (magnetite), with some magnetite framboids (top right of the cross section). Phosphate (darker grey than magnetite) are shown at the top of the cross section. Even darker grey carbonate occurs adjacent to the coarse Fe sulphide. The remaining darker grey is the phyllosilicate groundmass. Darkest grey features are OPs while void space is black. The Inset shows the coarse OP in the middle of the cross section enlarged with BSE images of 9 slices across its tomography volume. The first slice shows a single hollow nanoglobule morphology attached to a larger overall morphology containing a hollow lobe adjacent to another lobe filled by the lighter grey fibrous phyllosilicate-like material. **b** 3D render of the OP with its external morphology (left), and 2 cross sections through the render (middle and right). Purple is the organic segment and green is the internal fibrous phyllosilicate-like material. The right cross section deeper into the OP shows the adjacent hollow lobe. Scale bar in tomography inset and in **b** is 2 μm.

content is clearly higher[17], suggesting that diffuse OM is a mixture of both soluble and insoluble organic molecules[19]. Atomic Force Microscopy (AFM)-IR and Fourier Transform – Infra-red Spectroscopy (FT-IR) analyses on Ryugu and CC samples[25,33] have shed more light on the functional chemical variation between different phyllosilicate domains. Elevated fractions of C-H and C-O vibration bands identified by FT-IR within phyllosilicate[34] are consistent with the aliphatic and carboxylic-rich composition of diffuse OM identified by C-XANES[18]. Fine and coarse phyllosilicate domain boundaries may have localised soluble organic molecules[34], possibly recording the progression of aqueous fluid evolution that redistributed OM during phyllosilicate formation on Ryugu[34]. Rare occurrences of organic inclusions in minerals such as pyrrhotite[35] could also account for some of the SOM.

The characterisation of OM in situ in a grain from the first TAG site, A0083, is reported here, focusing on its morphology, distribution, and functional chemical variation. The novel application of FIB-SEM tomography coordinated with STXM and TEM is employed, developing on more traditionally used coordinated FIB-SEM-STXM-TEM techniques. A sample preparation method has also been designed for minimal mechanical damage to the grain, having important implications for future small particle preparation and analyses involving, e.g., Ryugu and Bennu. By comparing with CI, C1 and less altered CCs, we show that OM on Ryugu evolved into aromatic-rich OPs and aromatic poor, aliphatic/carboxyl-rich diffuse OM by aqueous alteration, similar to those found in type C1 planetary materials. We also report the discovery of OPs encapsulating silicate material (probably phyllosilicate) via FIB-SEM tomography, and discuss the implications this has on organic evolution on carbonaceous asteroids and their delivery to Earth.

## Results

Grain A0083 (Radegast here on in) is a 1.3 by 1.7 mm grain from Chamber A of the *Hayabusa-2* collector first TAG site. Radegast was prepared and analysed taking a unique approach of minimal damage and material loss for characterisation (methods section). The grain displays a mineralogy similar to grains from both chambers of the sample catcher, composed of coarse hexagonal Fe-sulphides (pyrrhotite) from the grain surface, widespread Fe-oxides (magnetite framboids) (Fig. 1), carbonates and lesser phosphates in a phyllosilicate groundmass. This CI composition is consistent between samples from both TAG sites, implying the same type of extensive alteration recorded in them.

### FIB-SEM tomography

Working from the edge of the grain (green rectangle in Fig. 1), a - 60 × 70 × 25 μm volume was targeted for FIB-SEM tomography and lamella preparation of ultra-thin samples for STXM-TEM on the OM and the surrounding petrofabric (methods section). Micron to submicron OPs with morphology and functional chemistry consistent with IOM from

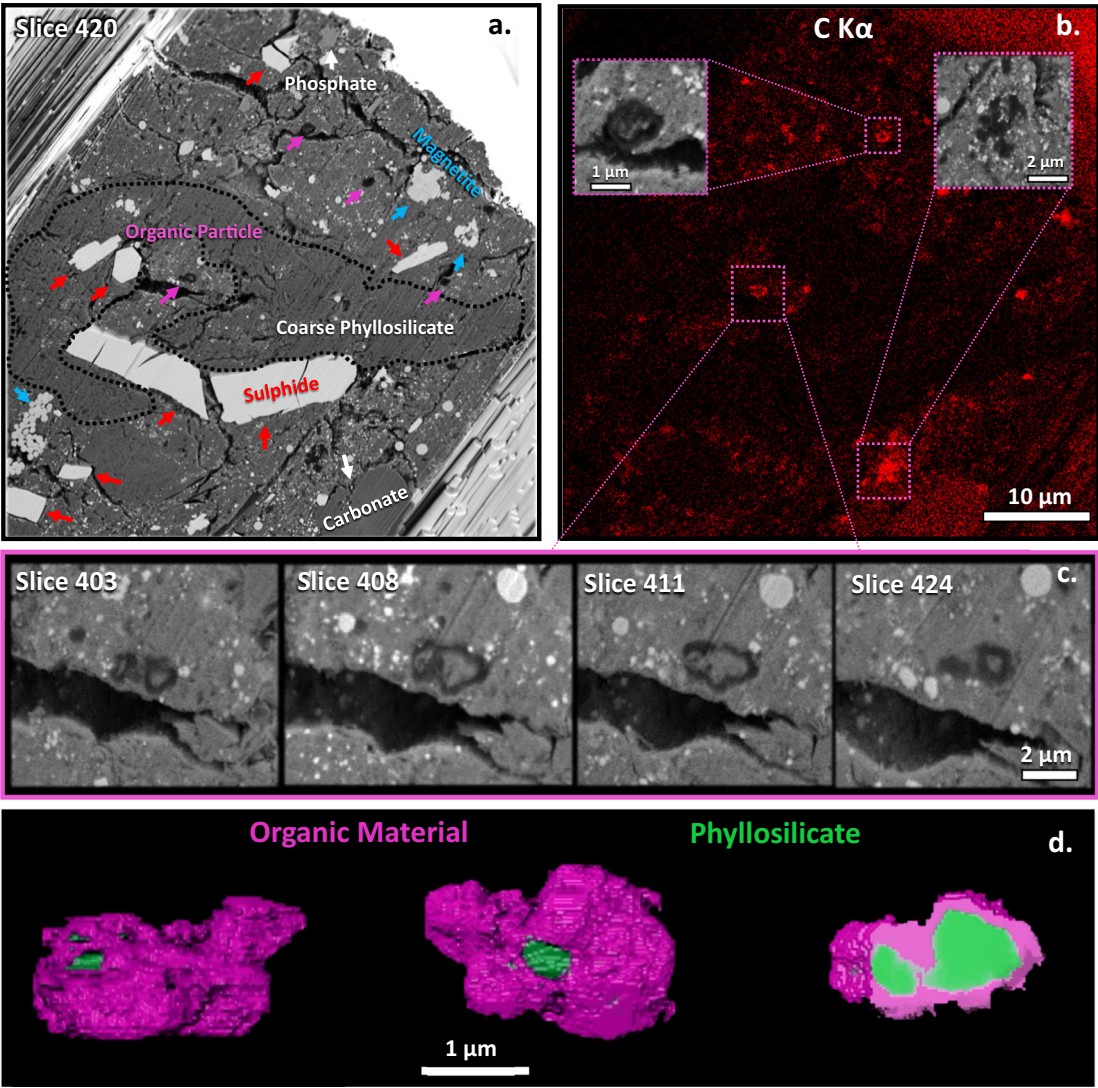

**Fig. 4 | Organic particles surrounding matrix material. a** BSE image of Slice 420 from the tomography showing another OP, microns below and deeper into the stack from the OP in Fig. 3. A coarse phyllosilicate domain is outlined (black dash). Red arrows are sulphides (probably pyrrhotite), blue is Fe-oxides (magnetite). Phosphate and carbonate are also shown, identified from the EDX maps (supplementary Fig. 2). **b** C Kα EDX map from the slice of the tomography set, shows the most intense regions corresponding to the OPs. Magnified insets show 2 of the coarsest OPs in the map. Inset on the left is a nanoglobule encapsulating matrix material. Right is a coarser grained OP with irregular shape but sub-particle compounded round nanoglobule-like morphologies. A carbonate grain is visible from the map at the bottom right and the BSE image in **a**. **c** four enlarged slices from the tomography of an OP shown in the C Kα EDX map in **b**. The slices show the OP made of adjacent lobes completely filled by the matrix-like material. **d** 3D Render of the OP in the middle of slice (purple OM and green silicate), in 2 orientations (left and middle). The phyllosilicate-like matrix material is shown cross cutting the OP. The sliced projection on the right shows the narrow organic wall surrounding the phyllosilicate-like material.

CCs are ubiquitous across the volume, as shown in the first tomography cross section from the edge of the grain (Fig. 2a, b). A ~ 5 μm coarse grained OP is shown at the top right of the cross section, vermicular shaped like polymers[36], and terminated by hollow and compounded nanoglobule morphologies (Fig. 2c). Most of the OPs in the cross sections are finer (modally sub 500 nm), displaying rounded nanoglobule morphologies. Some are however more irregularly shaped, as demonstrated by their 3D renders (Fig. 3). The rounded hollow and completely solid morphology of the particles combine to often form more irregularly shaped OPs (Figs. 3, 4). The coarse OP from the 1st tomography cross section in Fig. 2 shows that the hollow walled rounded features of the particles are composed of round solid ones (arrowed in Fig. 2c), suggesting formation controlled by surface tension effects similar to the formation of block polymers during polymerisation[36], rather than bubbling or cavitation involving the release of gases.

The coarsest OPs by slice and view tomography are shown in the supplementary movies. Supplementary Movie 1 is the entire volume. Slice and view of the OPs in Figs. 3 and 4 are in Supplementary Movies 2 and 3, respectively. The largest OP in Fig. 3 completely surrounds fibrous silicate material. EDX of the slices shows qualitatively an elemental composition similar to the surrounding phyllosilicate groundmass (Supplementary Fig. 1). 3D Rendering of the OP in Fig. 3 shows it to completely encapsulate the phyllosilicate-like phase inside of it (Supplementary Movie 4). The tomography slices also show the stark contrast between the morphology of the OP in different cross sections. One cross section shows just a single rounded hollow 200 nm nanoglobule morphology (slice 240 in Fig. 3a), whereas full tomography shows the OP to be an order of magnitude larger, displaying rounded lobes around the particle. Its tomography shows a hollow lobe on one side of the OP adjacent to a lobe completely surrounding the silicate (Fig. 3b). On the other hand, other OPs are completely filled

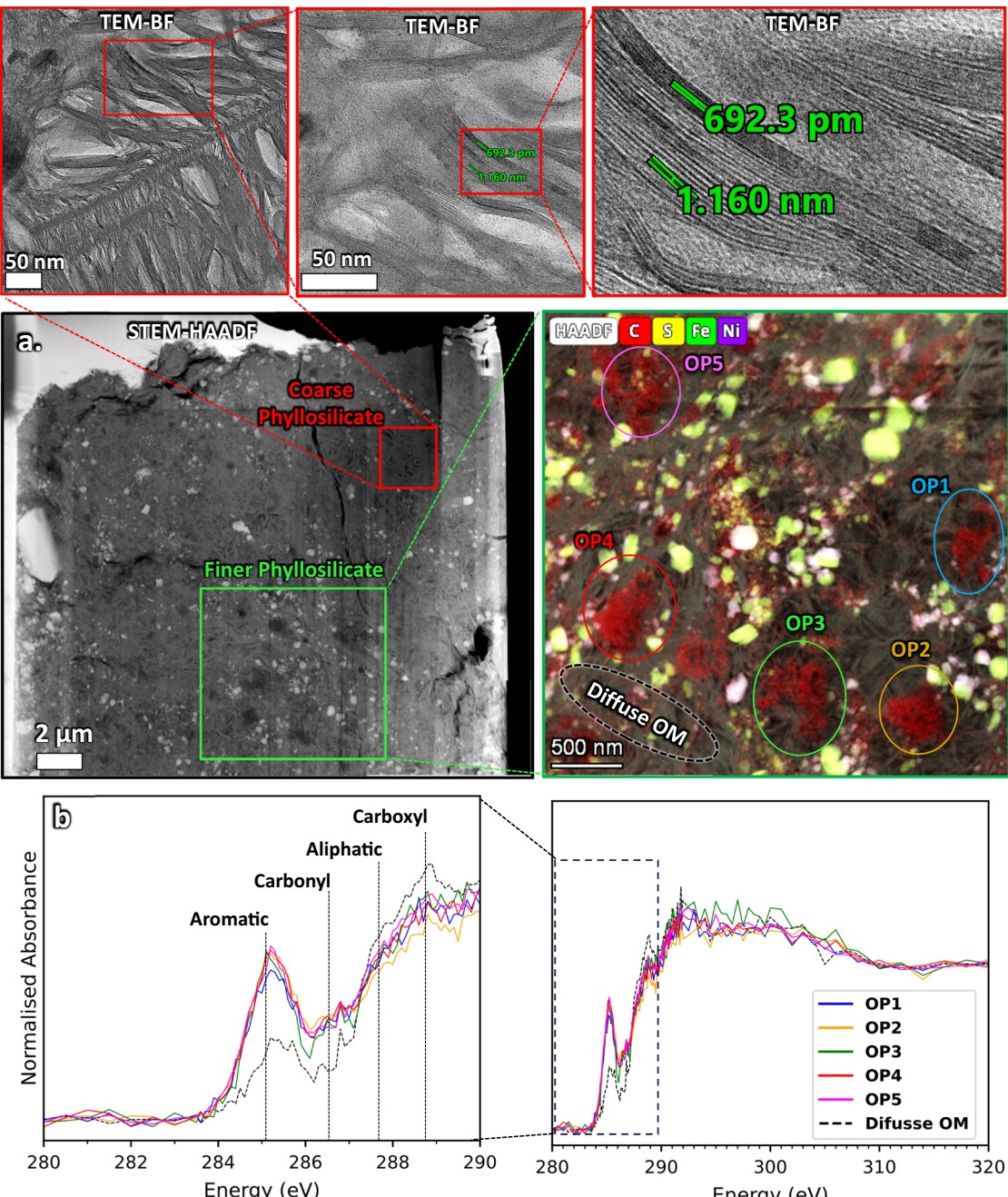

**Fig. 5 | STXM-TEM of lamella prepared with the TESCAN *LYRA3* (Ga$^+$ source).**
**a** HAADF image of entire lamella. Inset shows the region of STXM stack for XANES corresponding to where the dark grey OPs are mostly visible in the HAADF image. Right inset shows a coarse phyllosilicate domain magnified with HRTEM indicating 1:1 and 2:1 mix layered phyllosilicate. Magnified inset (left) is a STEM-EDX map showing 5 OPs in the phyllosilicate groundmass, intermixed with Fe Sulphides (Ni-

poor and Ni-rich). **b** C-XANES of all 5 OPs overlayed with adjacent ROI of diffuse OM (marked in SETM-EDX map in **a**). All of the OPs are the same position as 3-peak IOM – (aromatic C = C)-(carbonyl C = O)-(carboxyl COOH). The characteristic aromatic (C = C)-poorer and carboxylic (COOH)-richer diffuse OM, with additional shouldering X-ray absorption due to aliphatic (CH$_n$) functional chemistry, is overlayed with the OPs.

and crosscut by this fibrous, phyllosilicate-like material, and are composed of narrow organic walls, as shown in Fig. 4. Its 3D render is also shown in Supplementary Video 4. The OPs occur in the finer grained phyllosilicate domains as shown by the OP in Fig. 4. The texture of the surrounding coarse phyllosilicate domain is shown in Fig. 4 and Supplementary Fig. 2.

**STXM coordinated with TEM**
At the end of the tomography volume, two lamellae (one with a Ga$^+$ source and the other with Xe$^+$ PFIB) were extracted for C-XANES of the functional chemistry of the OM, followed by transmission electron

microscopy (TEM) for assessing the morphology of OM and its surrounding mineralogical context (Figs. 5, 6). A reference control sample of Ivuna was also prepared by ultra-microtome to assess the effects of FIB sample preparation on the OM (Fig. 7). Organic particles and diffuse OM are consistent between FIB and microtome samples (diffuse OM in microtome samples has not been reported before in any studies), implying that both Ga$^+$ (Fig. 5) and Xe$^+$ FIB (Fig. 6) preserve the functional chemistry of OPs and diffuse OM (Fig. 6). Carbon-XANES shows the OPs in both lamellae displaying aromatic-rich 3-peak characteristics similar to the 'aromatic' particles in Yabuta et al.[25]. Diffuse OM, aromatic-poorer and aliphatic-, carboxylic-richer than the OPs

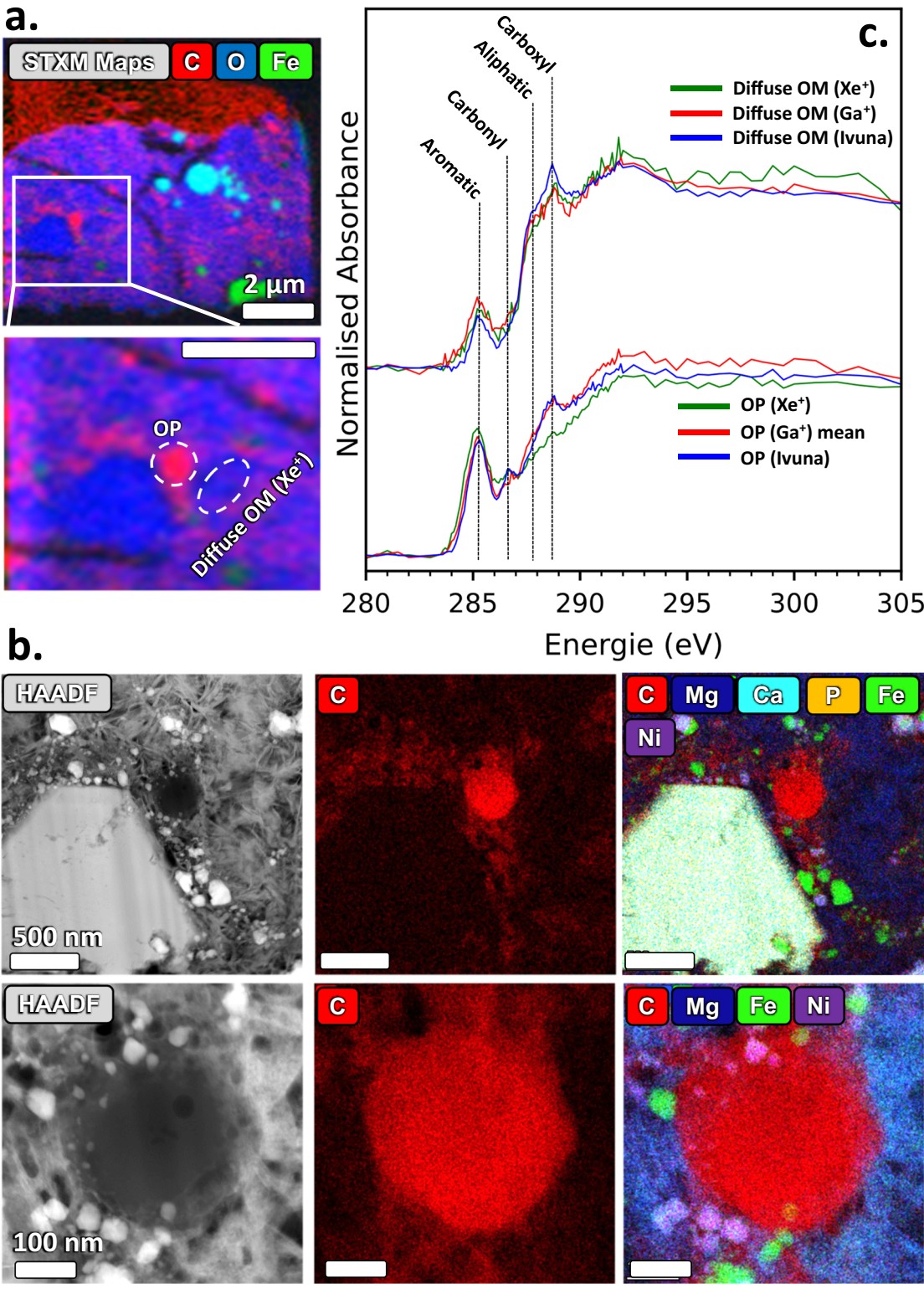

**Fig. 6 | STXM-TEM of lamella prepared with the TESCAN AMBER X (Xe⁺ source).**
**a** STXM RGB C, Fe, and O map. The XANES stack region is marked on the map with
the ROIs where the OP and diffuse OM spectra were taken. The green phase is an Fe-
sulphide and light blue are magneitte framboids. **b** HAADF image of the STXM
C-XANES stack region and associated STEM-EDX maps, showing an OP adjacent to a
carbonate apatite grain. The magnified HAADF image below it shows a round

OP ~200 nm across with a small hollow core and some irregular internal porosity.
**c** C-XANES of the OP overlayed with mean of the OPs in the Ga⁺ lamella and an OP
from an Ivuna microtome sample (Supplementary material Fig. 3). All OPs are
3-peak with the similar peak heights but low carboxyl 288.8 eV peak. Additional
XANES over the STXM stack are provided in Supplementary material Fig. 4, showing
the occurrence of diffuse OM in all regions of the silicate groundmass.

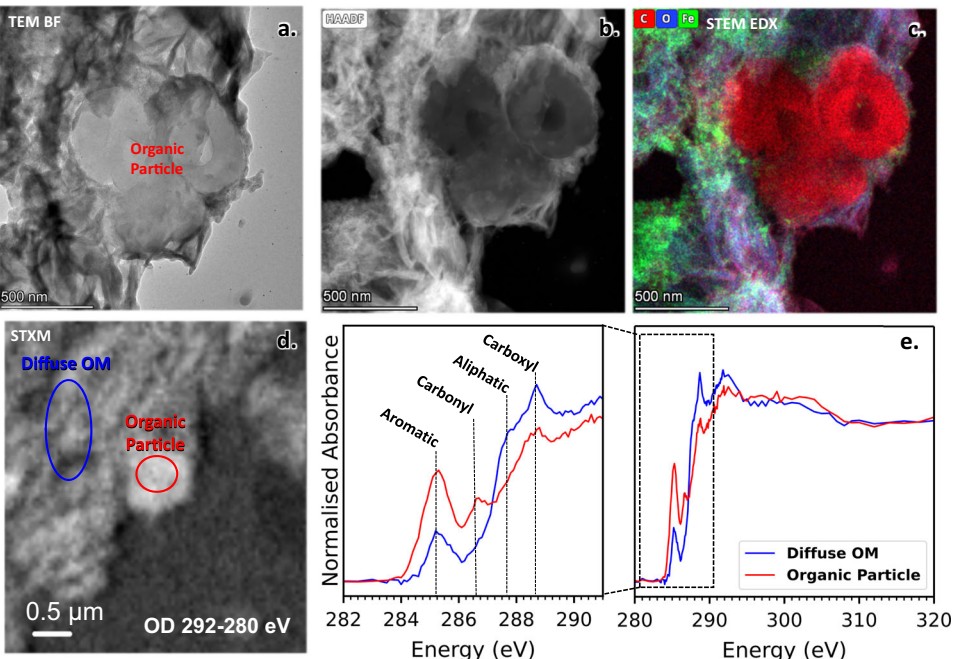

**Fig. 7 | STXM-TEM of Ivuna microtome sample. a** BF-TEM image of an OP adjacent to phyllosilicate. **b** STEM-HAADF. **c** STEM-EDX RGB map of same region. Red is C Kα, blue is O Kα and green Fe Kα. **d** STXM C map (292-280 eV OD). For each image, the brightest pixel intensity is taken as $I_0$ when calculating the OD. Approximate ROI regions are shown for the OP (red oval) and diffuse OM (black oval). **e** XANES of the OP with adjacent diffuse OM.

also occurs in both lamella in any region of the hydrated silicates which dominate the samples. Overlaying the C-XANES of the OPs in Radegast with OPs from a range of CCs (Fig. 8) shows its closest similarity with OPs in Ivuna and GRO 95577. The carbonyl peak absorption in Ryugu and type 1 can be ~50% lower than in IOM-like XANES of QUE 91177 (type 2/3) (Fig. 8). Aromatic-rich 3-peak OPs are in higher abundance in both the CR1 GRO 95577[24] and Orgueil[18] than in type 2/3 CCs more abundant 3-peak IOM-like XANES. Diffuse OM (aromatic-poorer and aliphatic/carboxylic-richer OM than OPs) in Radegast is also most similar to diffuse OM found in Ivuna (Fig. 7) and GRO 95577 (Fig. 8), with higher carboxyl/aromatic peak ratios than type 2/3. Figure 8 shows X-ray absorption in the aromatic (C = C) peak in diffuse OM of QUE 99177 up to ~100% higher than the type 1 CCs, and the carboxyl peak (COOR) at 288.8 eV in type 1 CCs up to ~40% higher than in QUE91177. Diffuse OM is also similar in both type 1 CCs and Ryugu, namely by being in coarse phyllosilicate rather than hydrated amorphous silicates or much finer polycrystalline phyllosilicate found in type 2/3 CCs.

SEM and TEM (Fig. 5, Supplementary Fig. 2) clearly shows coarse and finer domains of phyllosilicate. Diffuse OM occurs in both domains of phyllosilicate but is less concentrated in the coarse domains than the finer ones that are intermixed mostly with Fe and Fe,Ni-rich sulphides (Figs. 5, 6). HRTEM shows the coarse domains to be mixed layers of 1:1 and 2:1 phyllosilicate, consistent with previous studies[37] identified as interlayered serpentine and saponite.

## Discussion

Thousands of near-Earth asteroids have been identified, of which ~20% are carbonaceous C-type[38]. Both asteroids, Bennu and Ryugu, classified as carbonaceous Cb type from remote observations have shown to be of the most water- and organic-rich type of planetary material of CI chondrite composition. This means that at one stage in both of these asteroids' pasts, water was sufficient enough to have erased primary chondritic nebula components such as chondrules, CAIs, and amorphous silicates. Traces of primary mineralogy have been identified in Ryugu samples[39–41]. Possible micron-sized chondrule and CAI fragments are among the largest primary features remaining[39]. The

relatively small sizes of these asteroids, insufficient for ice to have melted through radioactive heating, coupled with dynamical modelling, suggest that they may have initially undergone alteration on larger planetesimals or even larger icy bodies[42]. Following assumed catastrophic impact, they would have reassembled into the asteroid rubble piles visible today[13]. Alternatively, heat sources for smaller icy bodies could have been provided, once drawn into the inner Solar System and sublimated as comets[43]. Main belt comets with similar sizes to, e.g., Ryugu and Bennu could also represent similar types of bodies as Ryugu and Bennu that became extinct over time.

During the time when this hydrothermal alteration took place on Ryugu[44], OM evolved into a composition more similar to C1 chondrites than type 3/2 CCs. Type 1 CCs have higher abundances of aromatic-rich 3-peak OPs and diffuse OM that is carboxylic-richer and aromatic-poorer than diffuse OM in type 3/2 CCs (i.e., they have higher carboxyl/aromatic ratios), implying the role played by alteration in determining both of these types of organic compositions. Even though CI (and Ryugu being of CI composition) is more primitive than other CC groups with a closer Solar composition of non-gaseous elements[45], alteration in type 1 CCs that are less primitive than CI, also account for the same organic composition[19]. Aromatic content in the XANES 285.4 eV peak can be high enough such that the adjacent 286.6 eV carbonyl peak is absent or virtually disappears due to the background of the aromatic peak, as shown in Orgueil[18] (Fig. 8a) and some other OPs from Ryugu[25]. This could be related to a process of maturation of the OM with alteration, likened to terrestrial shales on Earth[19,46]. Side branches and shorter chains are lost from the IOM during the most extensive alteration as recorded in Ryugu and type 1 CCs, enriching this aromatic fraction into 'highly aromatic' OPs in Ryugu, Orgueil[18] and the CR1 GRO 95577[19]. Their highly aromatic content show similarities with over mature terrestrial shale kerogen[46], unlike the more immature 'IOM-like' OPs in type 3/2 CCs[19]. Notably, within the more mildly metamorphosed CCs, unequilibrated ordinary chondrites (UOCs)[47] and the unique organic-rich clast in the Zag OC[48,49], aromatic-rich 2-peak OM is predominately found in these meteorites[47]. This implies that the intensified effects of parent body processing of IOM resulted in this highly aromatic-rich composition.

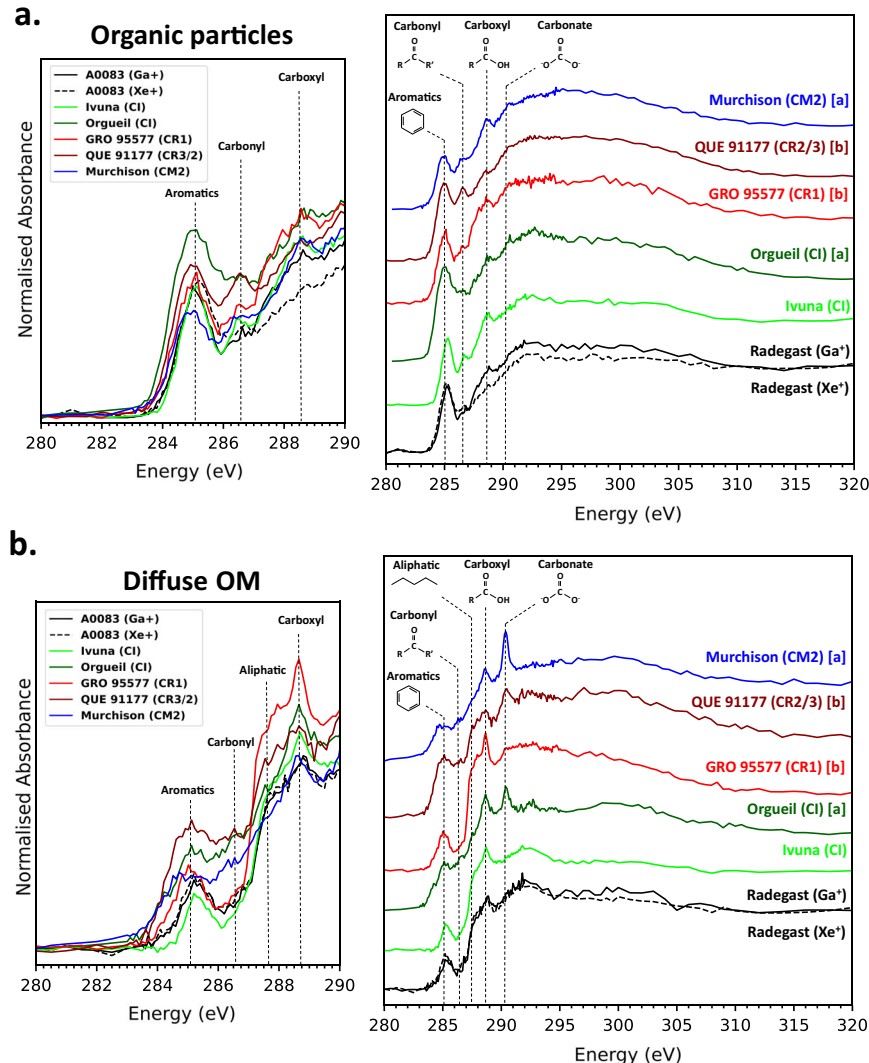

**Fig. 8 | XANES of OPs and Diffuse OM from A0083 compared with Ivuna (CI)** (Fig. 7), Orgueil (CI) and Murchison (CM2) (Le Guillou et al. [18][a]), and QUE 91177 (CR3/2) and GRO 95577 (CR1) (Changela et al. [19][b]). a Organic particles. The 3-peak OPs overlay with closest similarity to Ivuna and GRO 95577. They have lower carbonyl peak intensities relative to the aromatic peak than the XANES in QUE 91177 and Murchison where the aromatic/carbonyl peak ratios are lower – they are aromatic poorer than Ryugu particles and type 1 CCs. The most aromatic-rich particle is in Orgueil here. **b** Diffuse OM. XANES overlay of diffuse OM around the OPs shown in **a** from this and previous studies. A0083 diffuse OM is also most similar to Ivuna and GRO95577 in overall shape. The carboxylic/aromatic peaks ratios are higher in type 1 CCs and A0083 than type 3/2 CCs.

In parallel, diffuse OM within phyllosilicate is enriched in carboxyl and aliphatic functional chemistry. Cracking products of alteration of IOM leading to diffuse OM are consistent with increasing aromatic content of IOM by extensive alteration, as found in Ryugu[19]. Alternatively, diffuse OM could be SOM preserved within hydrated silicates after its polymerisation into OPs[28]. Oxidation of IOM could also account for its formation. Diffuse OM C-XANES in Ivuna having a slightly higher carboxyl peak than diffuse OM in Radegast (Fig. 8) could reflect terrestrial oxidation of OM, coinciding with the oxidising terrestrial environment forming terrestrial nano ferrihydrite and sulphates in CI[50]. Along the same lines, carboxyl fractions in type 1 CCs and Ryugu that are higher than in type 2/3 diffuse OM could also reflect oxidation of SOM or IOM on Ryugu. Constraints on models of the hydrothermal fluid conditions (e.g. Eh, pH) on Ryugu could possibly either support or rule out some of these formation scenarios. Ito et al.[37] reported a unique type of aliphatic ($CH_n$) – carboxylic (COOR) rich diffuse OM (all previous reports of diffuse OM are aromatic-poorer and carboxylic-richer than OPs[18,19]) in a Ryugu FIB lamella from the 2nd

chamber (targeted ejecta Ryugu material). Aliphatic-rich diffuse OM has never been reported in any CCs via C-XANES and could represent a unique organic component possibly preserved within phyllosilicate at the lower depths of the asteroid. Their C-XANES spectra are consistent with hydrocarbons, which form under more reducing conditions than the most abundantly found aromatic-poor and carboxylic-rich diffuse OM.

The discovery of micron-sized OPs enclosing fibrous phyllosilicate-like material suggests several possibilities: either the organic matter polymerised around the initial mineralogy, subsequently transforming into phyllosilicate; IOM polymerised after the formation of phyllosilicate; or siliceous fluids contained within pre-existing hollow OPs led to the formation of phyllosilicate within the OPs. SEM-EDX shows that qualitatively, the phyllosilicate inside the OPs is the same composition as the surrounding groundmass which most likely formed by the replacement of primary mineralogy (Supplementary Fig. 1). Observations have been made of OPs surrounding amorphous silicate grains in the CR2/3 QUE 91177[19]. Ito et al.[37] also

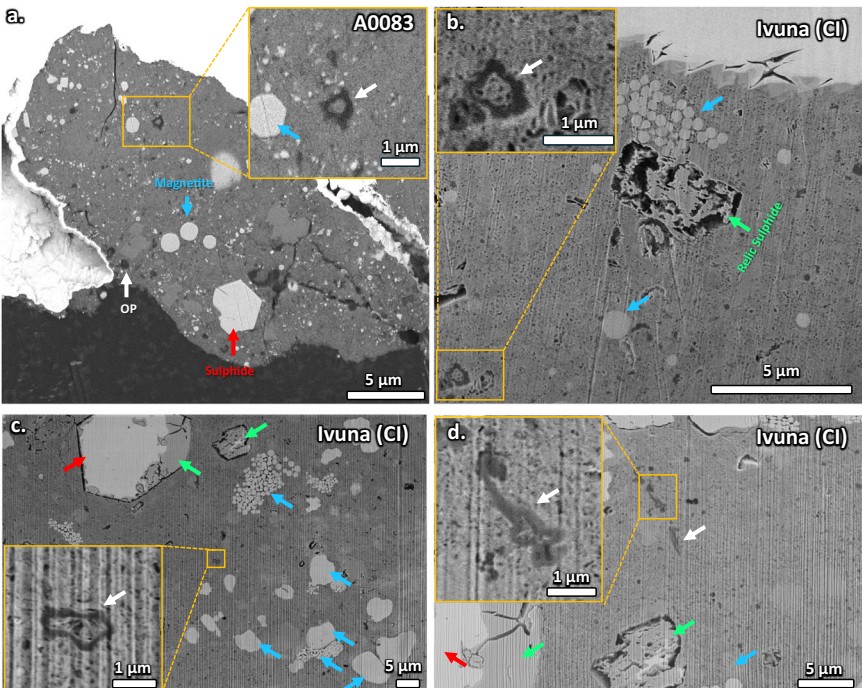

**Fig. 9 | Additional examples of OPs encapsulating matrix material in A0083 and Ivuna FIB-SEM cross sections. a** BSE image of a cross section of a fragment of A0083 with the AMBER X extracted from the facility-to-facility transfer container (FFTC) sapphire dish. Inset shows brighter inorganic material probably surrounded by darker OP. **b, c** SE and BSE images from a cross sections of Ivuna with the AMBER X and Quanta 3D, respectively. The insets show porous phyllosilicate-like material encapsulated by the dark grey organic walls of the OPs. **d** Ivuna with the Quanta 3D. OPs with vermicular morphology and nanoglobule sub particle morphology at the end of the worm, which also encapsulates the phyllosilicate-like material. Some other OPs are shown with white arrows. Red arrows show sulphides. Relic sulphides occur (green arrows) in Ivuna, and the altered edge of the hexagonal sulphide in **c**, after terrestrial exposure. Blue arrows are magnetite framboids. Note the curtaining effect in Ivuna in **c** and **d** by cross sectioning without stage rocking on the Quanta 3D.

reported a nanoglobule surrounding amorphous silicate within a Ryugu sample[37]. Mineralogy cannot be found inside nanoglobules IOM[23,24], implying that any encapsulated mineralogy was likely demineralised even when completely surrounded by IOM. This means that a primary silicate grain surrounded by IOM could still be susceptible to aqueous alteration, perhaps through nanometre sized cavities in the walls of the OPs, although the exact process is unclear. The major phase in lesser altered CC matrix is hydrated amorphous silicate, which would have contributed mostly to their replacement by phyllosilicate during hydrothermal alteration[51]. Amorphous silicate in type 2/3 or phyllosilicate in type 1 CCs are always hydrated[52] and contain diffuse OM[19]. Any ROI over hydrated silicate in Radgast shows the presence of diffuse OM (Supplementary Fig. 3). Le Guillou et al.[18] first proposed that SOM was within diffuse OM. Hydrated silicates can retain short organic molecules within their sheet silicate layers[53]. Organic adsorption at hydrated amorphous silicate surfaces by aqueous fluids[54] could facilitate the incorporation of soluble organic molecules within the primary silicate. Any OPs containing hydrated silicate such as the novel OPs in Ryugu, Ivuna and type 1 CCs will therefore contain diffuse OM and probably SOM as well. This feature of OPs is not uncommon in type 1 CCs as shown by random cross sections from a Radgast fragment (Fig. 9a) and grains of Ivuna (Fig. 9b–d).

Approximately 14 tons of extraterrestrial material is delivered to Earth per day[55]. The delivery of biologically relevant molecules to Earth within phyllosilicate and amorphous silicates would have been in higher abundance in the early Earth than today[56], which is mostly delivered as fine grain dust fragments of asteroids and comets. The most friable material being from CCs and in particular perhaps CI-like material raises a question about the preservation of biologically relevant materials delivered to Earth's surface. A clearer picture is now emerging on the lower abundance of CI chondrites surviving

atmospheric entry to Earth's surface. The survival of organic material upon impact with planetary atmospheres is dependent on the nature of biologically relevant molecules delivered to the surfaces of terrestrial planets[57,58]. By incorporating constraints on the and physicochemical properties and abundances of both organic bearing phyllosilicate and polymeric organic particles encapsulating phyllosilicate from early planetary bodies, a clearer account of the delivery of biologically relevant materials to the surfaces of early terrestrial planets can be made.

## Methods
### Samples and preparation
A0083 (Radgast) was prepared and analysed taking an approach of minimal mechanical damage to the grain for preparation and characterisation. Radgast was prepared at CEITEC BUT's class 100 cleanroom. The grain has to be secured to prevent any microscopic movements up to a Stage tilt -60° for FIB-SEM. Being an organic-rich, hydrated silicate rock mms in size, charging has to also be avoided for SEM imaging and FIB milling during tomography and ultrathin lamella preparation. A ~ 2 mm diameter divot was made on a clean strip of Au and mounted on an Al stub for Radgast to be placed on. The grain was picked up with vacuum tweezers and placed on the Au divot. An ~1.5 mm circular hole in clean Al foil was placed over the top of Radgast and fastened at the edge of the Al stub using double sided carbon sticky tape. The grain protrudes out of the hole and is secured by the tension of the foil around the edge of the grain. Radgast was left uncoated, with charging effects reduced via contact with the foil and Au base. This approach has the advantage that the grain can be turned over by lifting the foil for any future studies whilst leaving it in-tact, without any applied adhesives or coatings (Supplementary Figs. 4, 5).

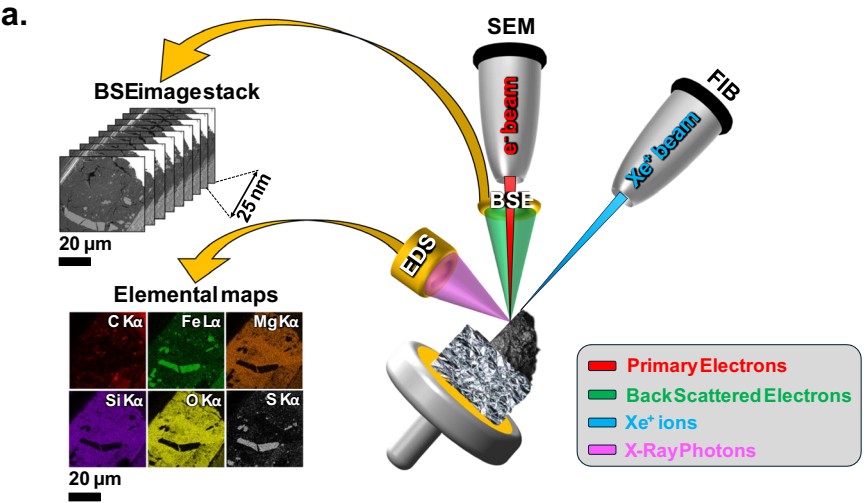

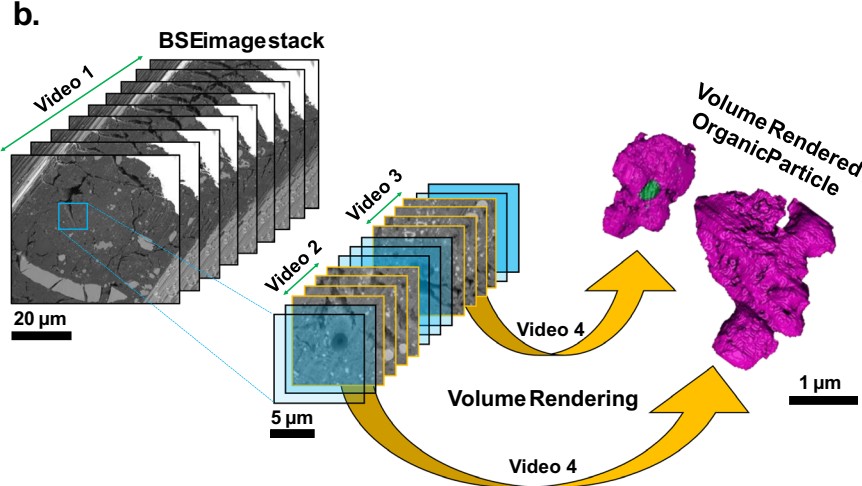

**Fig. 10 | Schematic of FIB-SEM tomography prior to lamella preparation from the edge of grain A0083 (Radegast). a** The sample is oriented perpendicular to the Xe⁺ ion beam in the AMBER X. BSE images are taken from each FIB polished cross section, with a beam shift of 25 nm into the volume for each slice. For every 4 slices, an EDX map is made at low kV (5 kV) to minimise the interaction volume. **b** From the tomography data stack, organic particles were segmented and rendered from the volume.

Microtome samples of Ivuna were also analysed (Fig. 7) as a control to compare the effects of FIB sample preparation on the OM. A sub mm grain of Ivuna was embedded in molten S for ultramicrotome[59]. An approximate mm-sized grain of Ivuna was dropped in molten S kept at higher temperatures at -180 °C, and kept refrigerated for 30 mins in order to have embedded in polymerised S. Samples were then sliced using a Leica UC7 at CEITEC-Muni and deposited on SiO grids. In addition, small grains of Ivuna we also placed on a carbon sticky tape and coated with 5 nm Au for FIB-SEM cross section analyses. The friable nature of carbonaceous asteroid material means that tiny fragments are likely to fall off of the grains by contact with harder material. A few micron sized fragments of Radegast on the sapphire dish in the facility-to-facility transfer container were collected by dabbing a carbon sticky on the dish after the extraction of the main grain. Both Radegast fragments and Ivuna grains were analysed by FIB-SEM cross sectioning with the TESCAN AMBER X (Fig. 9a, b) at 30 kV and 1 nA final polishing currents. Additional FIB-SEM cross sectioning was performed on Ivuna with the FEI Quanta 3D at the University of Leicester's Advanced Microscopy Centre (Fig. 9c, d) with the same accelerating voltages and currents, but without stage rocking that was used with the AMBER X.

## FIB-SEM

The TESCAN AMBER X and LYRA3 at TESCAN Group, Brno and CEITEC BUT, Brno respectively, were used to compare the effects of Xe⁺ PFIB with Ga⁺ FIB lamella preparation on the STXM measurements of OM. Two ~2 μm thick FIB sections were extracted and attached to Cu Omniprobe grids using the TESCAN AMBER X and LYRA3, respectively. Platinum strips ~20 μm long, 2 μm wide, and 2 μm thick were deposited on the samples to minimise ion beam damage during sample preparation. Approximately 100 nm thick sections were prepared by FB-SEM. For initial extraction from the grain, ion beam milling was carried out at a Ga⁺ (LYRA3) and Xe⁺ (AMBER X) ion beam voltage of 30 kV and beam currents ranging from 500 pA to 1 nA. Thin sections, 1–2 μm in thickness, were lifted out in situ using a micromanipulator and were transferred to a Cu TEM half grid for final ion milling. Ion currents from 300 pA to 30 pA were used for the final stages of thinning to achieve a thickness of about 100 nm. To minimise radiation damages due to the electron beam, notably the formation of a polycarbonate XANES peak at ~290.3 eV[60] (Supplementary Fig. 6), imaging using secondary electrons was kept to a minimum and was carried out at low voltage (5 kV and 2 kV at final thickness) and low beam current (30 pA). In addition, fast scan integration mode was used for final snap shots after final

polishing of the lamella at 2 kV. SEM surveys of the grain were also performed using the Hitachi S-4800 field emission gun SEM at the J'Heyrovksy Institute of Physical Chemistry, Czech Academy of Sciences.

## FIB-SEM tomography

FIB-SEM tomography was performed on an ~60 × 65 × 20 μm volume on the edge of Radegast using the TESCAN AMBER X and TESCAN 3D software. This was coordinated with lamella lift-out at the end of the volume with the AMBER X and TESCAN LYRA3 using $Xe^+$ and $Ga^+$ FIB ion sources, respectively. An ~60 × 30 × 5 μm protective Pt cap was initially deposited on the tomography and lamella lift out region. A drift correction marker was milled on Pt behind the cap for automated alignment corrections made by the TESCAN 3D tomography software. 25 nm beam shifts were set producing a tomography stack 25 × 25 × 25 nm/pix. Stage rocking on the AMBER X ± 7° was applied for the automated cross sectioning at 10 nA, minimising FIB curtaining for a smooth final cross section for each slice. 5 kV EDX maps were acquired every 4 slices for low kV elemental analysis using an attached Oxford Instruments Ultim 170 Silicon Drift Detector. This enabled the collection of C Kα, Fe Lα, Si Kα, S Kα, Mg Kα, P Kα and O Kα photons up to the overload voltage of Ca Kα. Tomography stacks were aligned using Microscope Image Browser[61] (MIB) and TESCAN 3D. Segmentation of organic particles was performed using MIB, and renders were made using Thermo Fisher Amira software (Fig. 10).

## STXM

STXM measurements were made at the Photon Factory KEK Beamline BL 19 A. Carbon maps on the lamella and Ivuna microtome samples were made by taking optical density (OD) images using the Beer-Lambert Law, at 292 eV OD (brightest pixel is $I_0$) and subtracting background at 280 eV OD (brightest pixel for $I_0$). Fe and O maps were made by 709–705 and 539–525 eV, respectively. From the C maps, regions were selected for the XANES maps (Figs. 5 and 6). Multiple energy X-ray absorption images from 280–320 eV were collected to form stacks, with 0.1 eV step for 3 ms each. XANES spectra were extracted using AXIS-2000 software[62], with background subtraction pre-edge 280–284 eV and normalisation to the post edge from and 315–320 eV using ATHENA[63] X-ray reduction software.

## TEM

Transmission electron microscopy on the FIB lamella and Ivuna microtome sample was performed using the Thermo Fisher Talos F200C equipped with a Bruker EDX at CEITEC-MUNI. Low magnification HAADF imaging was done by HAADF (High-Annular Angle dark-field) in STEM mode followed by higher resolution HAADF-EDX mapping coordinating with the XANES stacks regions by STXM. High resolution TEM of phyllosilicate was performed with Bright Field (BF) under standard operating conditions at 200 kV.

## Data availability

All raw and processed data is available upon suitable request to corresponding author HGC.

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

## Acknowledgements

We are very grateful to JAXA for the 1st AO allocation of Hayabusa-2 samples. This work is a part of a research series funded by the grant no. 21-11366S of the Czech Science Foundation and the Technology Agency of the Czech Republic (TAČR, NCK) under the grant TN02000009/07 FREYA. The authors acknowledge the assistance provided by the Advanced Multiscale Materials for Key Enabling Technologies project, supported by the Ministry of Education, Youth, and Sports of the Czech Republic. Project No. CZ.02.01.01/00/ 22_008/0004558, co-funded by the European Union. We acknowl-edge CF CryoEM of CIISB, Instruct-CZ Centre, supported by MEYS CR (LM2023042) and European Regional Development Fund-Project UP CIISB (No. CZ.02.1.01/0.0/0.0/18_046/0015974). We also acknowl-edge CzechNanoLab Research Infrastructure supported by MEYS CR (LM2023051). YK was supported by Japan Society for the Promotion of Science KAKENHI (grant number JP21I18648, JP21H00036, JP23H01286, and JP23K17700). Experiments at BL-19A in Photon Factory (PF) were approved by PF-PAC (2022G587 and 2018S1-001). JCB was supported by the School of Physics and Astronomy, University of Leicester. Special thanks to Takis Theodossiou of the Greek Meteorite Museum, Athens, for the curation of the Ivuna meteorite.

## Author contributions

H.G.C., Y.K., L.P., E.C., J.C.B., L.N. and M.F. were involved with writing and editing the manuscript text. R.N., T.S., J.M., P.K., Z.H., R.H., T.S., S.Y., Y.T., K.S., H.T. and D.Z. were involved with data accumulation, data processing, and figure generation for the manuscript. V.P., A.N., T.Y. and T.S. were involved with curation and handling, and sample preparation at J'Heyrovsky Institute, CEITEC-Nano and JAXA.

## Competing interests

The authors declare no competing interests.
