## [Peer Review File · Nature Communications]

The evolution of organic material on Asteroid 162173 Ryugu and its delivery to EarthREVIEWER COMMENTS

Reviewer #1 (Remarks to the Author):

Review of manuscript entitled "The evolution of organic material on Asteroid 162173 Ryugu and its delivery to Earth" by Changela et al., submitted to Nature Communications

Using various laboratory techniques, the authors characterized the organic material incorporated in a grain sampled in situ from asteroid Ryugu by the Hayabusa2 mission. The manuscript is certainly of interest to the general readership. However, some modifications must be made before this manuscript can be considered for publication in Nature Communications.

Major comments:

- 1) There are various language-related issues. Some sentences are hard to understand. I highly recommend that an English native speaker proofreads this manuscript before resubmission.
- 2) The authors should clearly separate between 1. what is already known about asteroid Ryugu and its organic material and 2. what their work adds to the current state of knowledge. Please restructure/rewrite the manuscript, most importantly the Discussion section, accordingly.

Minor comments:

Abstract

Also here, can you please make clear what your work in this paper adds to the current state of knowledge? For example, you could add a sentence like "Here, we present..."

Line 23: What does "solvent soluble" mean? Is the organic material soluble in water, organic solvents or both?

Introduction

Line 31: The first sentence is a bit confusing. We only know about life on Earth. Do you think that prebiotic chemistry on early-Venus led to life on Earth? Please explain or rephrase.

Line 35: I suggest replacing "Minor" with "Pristine".

Lines 39-40: I don't understand this sentence. Do you mean "Deducing" instead of "Reducing"?

Lines 54-55: Bennu grains are also rich in phosphorus, see <https://news.arizona.edu/story/sweating-small-stuff-uarizona-scientists-have-begun-study-samples-asteroid-bennu>

Lines 57-59: I have trouble understanding this sentence.

Line 63: The abbreviations seem misplaced here. If SOM refers to Soluble Organic Matter and IOM refers to Insoluble Organic Matter, then clearly mention this.

Lines 94-95: Again, it's hard to understand this sentence. It seems that "originated from" could be deleted.

Line 104: clear -> clearly

Line 107: has -> have

Lines 107-108: examinations both in IOM residues using Raman and -> examinations of IOM residues using both Raman and

Line 116: Who reported about a nanoglobule surrounding amorphous silicate?

Line 126: Please mention which is the "first TAG".

Lines 129-134: It seems that you are already summarizing your results. This should be moved to the Discussion section of the paper.

Results

General: How much of the analyzed grain is made of organic material? How does this compare to other grains from Ryugu (and Bennu)?

Line 141: Can you be more specific? What carbonates (calcite, dolomite, something else) and phosphates (Mg-phosphates, Fe-phosphates, Na-phosphates, something else) did you find?

Line 147: How thin is "ultra-thin"? How many of these ultra-thin slices have you prepared/analyzed? Can you please add this information here or in the Methods section?

Line 160: I tried with three different software tools, but I could not open Video 1. All other videos worked fine.

Discussion

Lines 206-209: Please split this sentence.

Lines 224-227: Hard to understand. Can you please split this sentence?

Line 231: "Oxidation of IOM could also account for its formation." – Formation of what?

Line 235: Can you make constraints on the hydrothermal fluid conditions based on you results?

Methods

You mention that you performed FIB-SEM analysis in Brno. Can you please also tell the reader where you have performed the STXM and the TEM analysis?

References

Reference 8 refers to a preprint. Please replace this reference with the published paper:

<https://www.nature.com/articles/s41550-021-01550-6>

Figures

Figure 1, Line 563: Do you know what that ~150 μm spherical feature is?

Figure 5, Line 656: (c) -> (b)

Supplementary Material

A reference to Supplementary Material Figure 1 is missing in the main text.

Reviewer #2 (Remarks to the Author):

Review on manuscript "The evolution of organic material on Asteroid 162173 Ryugu and its delivery to

Earth"

by Dr. Changela and colleagues

The manuscript discusses some very interesting insights gained from the analysis of samples returned from the carbonaceous asteroid 162173 Ryugu. These samples, classified as CI (Ivuna type), are rich in water and organics, primarily composed of phyllosilicate that encapsulates micron to submicron macromolecular organic particles, known as insoluble organic matter. It is observed that the insoluble organic material on Ryugu has undergone alteration, resulting in organic particles that are richer in aromatic compounds compared to those found in less altered carbonaceous chondrite matrices, aligning with characteristics of CI and type 1 carbonaceous chondrites. Additionally, Ryugu contains diffuse organic material, which is aliphatic-, carboxylic-rich, and aromatic-poor, suggesting a mix of acid insoluble and solvent soluble organic molecules. This diffuse organic matter, with its distinct carboxyl/aromatic ratios, underscores the influence of aqueous alteration in its formation. Furthermore, some organic particles have evolved to encapsulate the phyllosilicate, indicating that soluble organic-bearing diffuse organic matter is contained within by aqueous alteration on Ryugu. This suggests that Earth has been, and continues to be, receiving micron-sized polymeric organic objects that encapsulate biologically relevant molecules, providing insights into the early Solar System's prebiotic evolution.

I like the topic and the results from a very successful mission very much. The manuscript may be publishable in principle. In order to focus the text and story I would include some pictures in the main text - not all of the figures should be in the SI only. That distracts the reader. Also, I would focus a bit more on the main findings in the discussions part or in an outlook. I see very important and nice results that clearly deserve publication in Nat. Comm., but for the wide readership of the journal it has to be prepared a bit more to the point in the introduction (what, how, why) and at the end (who cares, and what are the implications of the current findings). Maybe a scheme would also help - also for the non-expert. I confirm that the authors already tried to provide a nice story but I believe that this could be improved.

So, my recommendation would be: publish after revision - and considering my points above.

The evolution of organic material on Asteroid 162173 Ryugu and its delivery to Earth

Response to Referees Changela et al. Date: 17th March 2024

Thank you for the thoughtful reviews which we believe have significantly improved the manuscript. Please find here the corrections made to the manuscript addressing the comments by the reviewers. Corrections related to comments are also shown in the highlighted yellow text in the manuscript.

Reviewer 1:

Abstract

Also here, can you please make clear what your work in this paper adds to the current state of knowledge? For example, you could add a sentence like "Here, we present..."

Line 23:

Added the sentence, 'By coordinating focused ion beam-scanning electron microscopy tomography with scanning transmission X-ray microscopy and transmission electron microscopy, we find that...'

In line 24, we also added, This challenges the view that the formation of aromatic-rich insoluble organic matter is pre accretionary, based on the recent interpretations of Yabuta et al. (2023) Science.

Line 23: What does "solvent soluble" mean? Is the organic material soluble in water, organic solvents or both?

Because diffuse OM could be soluble to water and/or organic solvents, we added, 'water/organic soluble.'

Line 31: The first sentence is a bit confusing. We only know about life on Earth. Do you think that prebiotic chemistry on early-Venus led to life on Earth? Please explain or rephrase.

We state that prebiotic chemistry MAY have led to life on those planets. This is why they are being explored for extant/extinct life.

Line 35: I suggest replacing "Minor" with "Pristine".

Because 'pristine' might be a more relevant description of type 3 unaltered bodies, we suggest replacing with 'Planetesimals'

Lines 39-40: I don't understand this sentence. Do you mean "Deducing" instead of "Reducing"?

Yes we meant from a reductionist approach. We suggest replacing with, 'Unravelling' instead.

Lines 54-55: Bennu grains are also rich in phosphorus,

Added phosphates.

Lines 57-59: I have trouble understanding this sentence.

Rephrased to, 'If Ryugu and Bennu are representative major carriers of carbonaceous asteroid material reaching Earth, the scarcity of CI chondrites in Earth's meteorite collection might be attributed to their relative fragility compared to other carbonaceous groups, possibly hindering their survival upon atmospheric entry.'

Line 63: The abbreviations seem misplaced here. If SOM refers to Soluble Organic Matter and IOM refers to Insoluble Organic Matter, then clearly mention this.

Line 67 replaced with, 'soluble organic matter (SOM) and insoluble organic matter (IOM)...'

Lines 94-95: Again, it's hard to understand this sentence. It seems that "originated from" could be deleted.

Deleted the word 'from,'²⁹ and added, 'possibly in non-mutually exclusive ways.'

Line 104: clear -> clearly

Replaced

Line 107: has -> have

Replaced

Lines 107-108: examinations both in IOM residues using Raman and -> examinations of IOM residues using both Raman and

Corrected

Line 116: Who reported about a nanoglobule surrounding amorphous silicate?

Ito et., (2023) is reported in the text.

Line 126: Please mention which is the "first TAG".

Added 'grain A0083.'

Lines 129-134: It seems that you are already summarizing your results. This should be moved to the Discussion section of the paper.

We attempted to summarise relevant findings from recent studies. However, we have edited the text here, trying to make it simpler to read, by splitting the original paragraph 107-124 and incorporating into the previous paragraph about diffuse OM from lines 236, and into the subsequent paragraph on the OPs encapsulating phyllosilicate in the discussion.

Results

General: How much of the analyzed grain is made of organic material? How does this compare to other grains from Ryugu (and Bennu)?

Added in the results line 143, 'Micron to submicron OPs with morphology and functional chemistry consistent with IOM from CCs.'

Line 141: Can you be more specific? What carbonates (calcite, dolomite, something else) and phosphates (Mg-phosphates, Fe-phosphates, Na-phosphates, something else) did you find?

Our FIB SEM-EDX maps are only up to 5 KV, meaning that it is difficult distinguishing between e.g. Ca- or Mg- rich carbonates, and the types of phosphate we identified.

Line 147: How thin is "ultra-thin"? How many of these ultra-thin slices have you prepared/analyzed? Can you please add this information here or in the Methods section?

By ultra thin we mean electron and soft X-ray transparent. We made 110 nm microtome samples and with the challenge of measuring the thicknesses of FIB lamella, interpreted ~100 nm lamella. Between Ivuna FIB sections and microsamples (our Ivuna FIB sample data is not reported here), OPs have very similar optical densities, implying similar thicknesses between FIB and microtome samples.

Line 160: I tried with three different software tools, but I could not open Video 1. All other videos worked fine.

Video 1 should work on media viewer but will try resubmitting a different format if need be.

Discussion

Lines 206-209: Please split this sentence.

Rephrased to, The relatively small sizes of these asteroids, insufficient for ice to have melted through radioactive heating, coupled with dynamical modelling, suggests they might have initially undergone alteration on larger planetesimals or even larger icy bodies.⁴² Following catastrophic impacts, they likely reassembled into the asteroid rubble piles visible today.¹³

Lines 224-227: Hard to understand. Can you please split this sentence?

Rephrased to, 'Notably, within the more mildly metamorphosed CCs, unequilibrated ordinary chondrites (UOCs)⁴⁷ and the unique organic-rich clast in the Zag OC^{48, 49}, aromatic-rich 2-peak OM is predominately found in these meteorites.⁴⁷ This implies that the intensified effects of parent body processing of IOM resulted in this highly aromatic-rich composition.'

Line 231: "Oxidation of IOM could also account for its formation." – Formation of what?

Diffuse OM in the previous sentence.

Line 235: Can you make constraints on the hydrothermal fluid conditions based on you results?

Our results show how the composition of the organic particles and diffuse OM compare with CCs. We included the sentence stating how, 'Constraints on models of the hydrothermal fluid conditions (e.g. Eh, pH) on Ryugu could possibly either support or rule out' some of the formation scenarios of diffuse OM in line 234.

Methods

You mention that you performed FIB-SEM analysis in Brno. Can you please also tell the reader where you have performed the STXM and the TEM analysis?

This is in the 'STXM' and 'TEM' section of the methods.

Figures

Figure 1, Line 563: Do you know what that ~150 μm spherical feature is?

At this stage we only know it to be a nodule in the grain, which requires further analysis. Included the word 'nodule' in the figure caption.

Figure 5, Line 656: (c) -> (b)
Corrected, thank you.

Supplementary Material

A reference to Supplementary Material Figure 1 is missing in the main text.

Included in line 156, 'EDX of the slices (supplementary Fig. 1) shows qualitatively an elemental composition similar to the surrounding phyllosilicate groundmass.'

Reviewer 2 and Reviewer 1 major comments

In order to focus the text and story I would include some pictures in the main text - not all of the figures could be in the SI only. That distracts the reader. Also, I would focus a bit more on the main findings in the discussions part or in an outlook. I see very important and nice results that clearly deserve publication in Nat. Comm., but for the wide readership of the journal it has to be prepared a bit more to the point in the introduction (what, how, why) and at the end (who cares, and what are the implications of the current findings). Maybe a scheme would also help - also for the non-expert. I confirm that the authors already tried to provide a nice story but I believe that this could be improved.

We ordered the discussion with the 1st paragraph on the origin of Ryugu, the 2nd on the origin of aromatic rich OPs, the 3rd on the formation of diffuse OM, the 4th on the discovered of phyllosilicate bearing OPs, and the final one on the delivery of the OPs bearing biologically relevant molecules to Earth. We incorporated the Ivuna data from the supplementary material into the main text, with some additional data for figure 9 showing some more phyllosilicate silicate bearing OPs in Ivuna. We have attempted to simplify the introduction by taking the original penultimate paragraph on previous studies and incorporated it into the previous paragraph of the introduction and into the discussion. As the FIB-SEM application is a novel technique, we have made a schematic figure included as figure 10 in the methods section. Some grammatical changes have also been made the text.

REVIEWERS' COMMENTS

Reviewer #1 (Remarks to the Author):

The authors properly addressed most of my comments. I would like to ask them for the following three minor changes prior to publication:

1) Please mention in your manuscript that your FIB SEM-EDX maps are only up to 5 KV, meaning that it is difficult distinguishing between e.g. Ca- or Mg- rich carbonates, and the types of phosphate you identified.

2) In the last paragraph of the Discussion section, you state that approx. one ton of extraterrestrial material would be delivered to Earth per day. But it is probably more than that. A study by Rojas et al. (2021) finds that about 5200 tons per year (~14 tons per day) hit Earth's surface:
<https://www.sciencedirect.com/science/article/pii/S0012821X21000534?via=ihub>

3) Please replace reference 8, which refers to a preprint, with the published paper (as already requested in my first review report): <https://www.nature.com/articles/s41550-021-01550-6>

Reviewer #2 (Remarks to the Author):

The authors have now answered my questions and took all (most of the) suggestions into account. The manuscript has now considerably improved. It is now acceptable for publication.